# Applications of Medical Digital Technologies for Noncommunicable Diseases for Follow-Up during the COVID-19 Pandemic

**DOI:** 10.3390/ijerph191912682

**Published:** 2022-10-04

**Authors:** Eman Sobhy Elsaid Hussein, Abdullah Mohammed Al-Shenqiti, Reda Mohamed El-Sayed Ramadan

**Affiliations:** 1Nursing Department, College of Applied Medical Sciences-Yanbu, Taibah University, Medina 42353, Saudi Arabia; 2Medical Surgical Nursing Department, Faculty of Nursing, Ain Shams University, Cairo 11566, Egypt; 3Centre for Rehabilitation Sciences, University of Manchester, Manchester M13 9PL, UK; 4Faculty of Medical Rehabilitation Sciences, Taibah University, Medina 42353, Saudi Arabia; 5Medical Surgical Nursing, College of Applied Medical Sciences, Shaqra University, Shaqra 15518, Saudi Arabia

**Keywords:** COVID-19, follow-up, medical digital technologies, noncommunicable diseases

## Abstract

Background: Noncommunicable chronic diseases (NCDs) are multifaceted, and the health implications of the COVID-19 pandemic are far-reaching, especially for NCDs. Physical distancing and quarantine can lead to the poor management of NCDs because the visual tracking of them has been replaced with medical digital technology, that is, smartphone apps. This study aimed to explore medical digital technology applications for NCDs for follow-up during the COVID-19 pandemic. Methods: The participants in this study were 400 adult patients with NCDs; they were selected by systematic random sampling. A descriptive cross-sectional design was used. The study was conducted in the outpatient department of Yanbu General Hospital and primary-care health centers in Yanbu Al-Baher, Al-Madinah Al-Munawwarah, in the Kingdom of Saudi Arabia. The tools used in this study were a structured questionnaire to collect the sociodemographic characteristics of the patients and their health history, an NCD questionnaire to assess follow-up of the patients during the COVID-19 pandemic, and a medical digital technologies questionnaire to explore the medical digital technology applications. Results: The mean age of the patients was 47.32 ± 14.362 years, and 62.8% of them were female and 372 were male. Of the patients, 69.2% and 57.5% had been diagnosed with diabetes mellitus and hypertension, respectively; 52.4% were followed up monthly, and 29.75% used medical digital technology applications such as Tabeby Online to monitor their health. Furthermore, 71.75% and 75.5% of the patients used the Sehhaty and Tawakkalna medical digital applications, respectively. Overall, 38.7% of the patients were satisfied with using medical digital technology applications used for follow-up during the COVID-19 pandemic. Conclusions: The study concluded that the services that use networks, smartphones, and medical digital technology applications on the Saudi Ministry of Health website and mobile applications to improve the quality of the health-care system, and that provide health services for noncommunicable or communicable diseases, are not effective. This is because the patients lack awareness of these services, with most of the chronic patients being elderly with lower levels of education and computer literacy.

## 1. Introduction

The prevalence and impact of noncommunicable diseases (NCDs) are increasing at a remarkable rate and the burden of NCDs remains a global public health challenge, leading to high mortality and morbidity [1,2]. Currently, NCDs such as cardiovascular disease, diabetes, cancer, and chronic lung disease account for approximately 60% of all deaths worldwide, with 80% of these deaths occurring in developing countries [3]. Such deaths would be partially preventable if preventive and follow-up measures were implemented effectively to reduce the risk factors of NCDs [4].

The incidence and prevalence of NCDs, which account for 70% of all deaths in Saudi Arabia [5], show an alarming increase based on epidemiological data. With the aging and the urbanization of the population, chronic diseases and multimorbidity are expected to pose a major threat to public health in Saudi Arabia [6]. 

The World Health Organization (WHO) defines NCDs such as heart disease, stroke, cancer, diabetes, and chronic lung disease [7] as long-term chronic conditions resulting from a combination of genetic, physiological, behavioral, and environmental factors [8] Management of NCDs includes screening, early detection, and treatment, as well as access to palliative care for people in need. Essential and highly effective NCD interventions can be implemented as part of a primary health-care approach to strengthen early detection and timely treatment. Evidence shows that such interventions are an excellent economic investment because when they are offered to patients early, they can reduce the need for more expensive treatments [9,10].

Interventions against NCDs are essential to achieve the global target of a 25% relative reduction in the risk of premature mortality from NCDs by 2025. The sustainable development goals (SDGs) aim to reduce premature deaths from NCDs by one-third by 2030 [7]. Conventional care for NCDs in a large population can be difficult, as patients with chronic conditions require continuous monitoring and prolonged treatment [11].

The fight against COVID-19 has severely impacted services to prevent and treat cancer, diabetes, hypertension, and other NCDs that kill more than 40 million people each year, according to a new study published by the WHO [12]. During the COVID-19 pandemic, prevention and control of NCDs is important because they are high-risk factors for infection. In addition, some restrictive measures such as stay-at-home orders, physical distancing, and travel restrictions to curb the spread of infections in many countries may specifically affect people living with NCDs by limiting their activities, thereby limiting their ability to obtain healthy food and their access to preventive or health-promoting services [13].

The World Health Organization recognizes the potential of digital health interventions to achieve universal health coverage and ensure quality care for individuals through the use of mobile phones, web portals, or other digital tools [14]. Digital technologies are an essential component of and an enabler for sustainable health systems and universal health coverage. To reach their maximum potential, digital health initiatives must be part of the patient’s health needs and digital health ecosystem. They must be guided by a sound strategy that integrates leadership, financial, organizational, human, and technology resources and serve as the basis for an action plan that enables coordination among multiple stakeholders [15]. The use of cost-effective innovative eHealth interventions, such as mobile health (mHealth), can help improve the prevention and control of NCDs in disadvantaged populations [16]. 

The safest mode of follow-up for the delivery of medical services is virtual or telecare follow-up during the COVID-19 pandemic. It minimizes contact and the spread of viruses and allows timely follow-up by maintaining distance. Digital media such as WhatsApp and other teleconsultation software, for example, hospital information systems, are being used effectively. Many mobile applications are also popular in private practice for teleconsultation and telerehabilitation. After discharge from NCD treatment, telerehabilitation services in outpatient departments should provide consultations in clinics [17].

Medical prevention and health centers improve the efficiency of follow-up of people with NCDs through information technologies, including telephonic follow-up. Hence, they are an important resource in the implementation of high-risk and secondary prevention strategies [18]. 

Medical digital technologies, which can be used via smartphone apps, allow structured monitoring of health parameters and timely feedback [8]. The use of wearable health technologies, mobile phone apps, and home monitoring devices by patients with NCDs has become popular [19].

### 1.1. Magnitude of the Problem 

The impact of COVID-19 response actions on NCDs is multifaceted and includes physical distancing and quarantine, which may lead to poor management of behavioral risk factors for NCDs, including unhealthy diet and physical inactivity [13]. Follow-up of NCDs is important to reduce the risk of morbidity and preventable mortality, and to limit severe acute and chronic complications that carry an increased risk of COVID-19 adverse outcomes [20]. During the COVID-19 pandemic, the significance of connecting health-care services increased. Due to physical distancing, travel limitations, and the use of digital medical applications for follow-up NCDs, these strategic priorities have been demonstrated for patients with NCDs.

### 1.2. Aim of the Study

This study aimed to explore the applications of medical digital technologies for follow-up of patients with NCDs during the COVID-19 pandemic.

### 1.3. Research Questions 

-What are the methods of follow-up for patients with NCDs during the COVID-19 pandemic?-What are the applications of medical digital technologies in the follow-up of patients with NCDs during the COVID-19 pandemic?-What are the services of applications of medical digital technologies for follow-up of patients with NCDs during the COVID-19 pandemic?

## 2. Subjects and Methods

### 2.1. Research Design

As a statistical study based on self-compiled questionnaires, a descriptive cross-sectional design was utilized to evaluate the uses of medical digital technology for NCD follow-up during the COVID-19 pandemic.

### 2.2. Setting

The study was conducted in the outpatient department of Yanbu General Hospital and primary-care health centers (Al-Arbaeen, Al-Sumairi, Al-Suraif, Al-Sharqiya, Al-Bandar and G-16) in Yanbu Al-Baher, Al-Madinah Al-Munawwarah, Kingdom of Saudi Arabia.

### 2.3. Subjects

The participants of this study were 400 adults. The men (*n* = 149) and women (*n* = 251) were aged 18 years or more, with a mean ± SD of (47.32 ± 14.362) for both sexes. All participants enrolled in this study were diagnosed with an NCD and were outpatients. Participants were excluded from the study if they were suffering from a critical case, cognitive impairment, or alcohol or drug abuse.

### 2.4. Sampling Design

The sample of participants was systematically randomly selected from a list provided by the Medical Records Department.

### 2.5. Sample Size

The researchers calculated the number of participants from the target population of subjects using the statistical records of the Department of Health Affairs for primary health-care centers in Yanbu affiliated with the Ministry of Health and using Steven K. Thompson’s equation to calculate the sample size as follows:
$$n=\frac{N*P(1-P)}{\left[(N-1*(\frac{d^{2}}{Z^{2}})+P(1-P)\right]}$$
where *N* = population size (22,610), *Z* = confidence level at 95% (1.96), *d* = error proportion (0.05), *P* = probability (50%), and the result *n* = sample size (378).

### 2.6. Tools for Data Collection

The data were collected using the following tools:

Structured questionnaire: This was developed by the researchers in Arabic and consisted of two parts: The first part dealt with the sociodemographic characteristics of the patients, such as age, gender, occupation, marital status, educational attainment, financial status, and location. The second part was used to collect the health history of the patients, including medical diagnosis related to NCDs, duration of diseases, regular medication, family history, smoking, previous accidents, allergic conditions, duration of sleep, and passive smoking.

NCDs questionnaire: This was designed by the researchers in Arabic to assess the follow-up of patients during the COVID-19 pandemic. The first part of the questionnaire addressed the characteristics of follow-up for the patients (frequency of follow-up, pattern of follow-up, progress of health status during follow-up, desire to visit member of the health team during follow-up, going to follow up with a relative, specified treatment during follow-up, symptoms and signs known to the patient requiring hospital visit, instructions/education given during follow-up, and any change in medication during follow-up). The second part dealt with the methods of follow-up during the COVID-19 pandemic. A reliability test was performed where the Cronbach’s alpha value was equal to 0.982.

Medical digital technologies questionnaire: This was designed by the researchers in Arabic to explore the applications of medical digital technologies for the follow-up of NCDs. The first part consisted of three questions about the availability of medical digital technologies for the patients during the COVID-19 pandemic; the second part consisted of nine questions about the use of medical digital technologies applications; the third part consisted of five questions about the services; and the last part consisted of questions about the patients’ satisfaction with the use of medical digital technology applications during the COVID-19 pandemic. A reliability test was performed where the Cronbach’s alpha value was equal to 0.952.

#### 2.6.1. Fieldwork 

The fieldwork was carried out for six months, from March 2021 to August 2021 inclusive. The researchers collected a sample from six primary-care health centers and outpatients in Yanbu General Hospital, each comprising 58 patients selected from the patients list; 1 in every 10 patients was chosen. The sample was chosen during morning shifts. The period of data collection estimated for each patient ranged from ten to fifteen minutes after explaining the purpose of the study by using Google Forms. The data were collected in an Excel spreadsheet to enter into SPSS for data analysis.

#### 2.6.2. Validity and Reliability

Face validity and content validity were used to test validity. Face validity aimed to check the instruments for clarity, relevance, completeness, simplicity, and applicability; minor modifications were made. Content validity testing for all instruments was reviewed by five experts from the academic staff of the Medical-Surgical Nursing Department at the Faculty of Nursing, Ain Shams University, to ensure that the assessment instrument provided stable and consistent results over time. Reliability analysis was established with Cronbach’s alpha to determine how closely the items in all the tools were related.

#### 2.6.3. Pilot Study

The pilot study was conducted on five patients in order to check the clarity, relevance, and applicability of the tools and estimate the time required for interviewing a patient. Based on the opinion of a panel of five experts and the results of the pilot study, some statements were omitted or rephrased, and then the final forms were developed. The patients who were included in the pilot study were excluded from the study sample

#### 2.6.4. Data Analysis

Data entry and statistical analysis were performed using the SPSS 16.0 statistical software package. Results are presented in terms of frequency and percentage.

## 3. Results

Table 1 shows that the mean age of the patients was 47.32 ± 14.362 years, with a minimum of 18 years and a maximum of 85 years; two-thirds were females (62.8%) and most were married (70.7%). Regarding occupation, 31.5% were housewives and 27.5% were office workers. Regarding educational attainment, 30.2% were highly qualified; more than two-thirds (71.2%) had sufficient monthly income, and the majority (84%) lived in a city.

Table 2 shows that almost two-thirds of the patients were diagnosed with diabetes mellitus and hypertension, 69.2% and 57.5%, respectively. Regarding the duration of illness of the patients, 34.5% ranged between 5 and 10 years; most took regular medication, did not smoke, and had no history of accidents, with 81.5%, 75.8%, and 81.5%, respectively. Two-thirds of them had a family history of NCD, did not suffer from allergies, and had enough time to sleep, with 66%, 68.2%, and 63.5%, respectively. More than half (54.5%) of the patients did not live with a smoker within or outside the family.

Table 3 shows that more than half of the patients reported monthly follow-up, monthly medication, no follow-up with a relative, and change in medication during follow-up, with 52.4%, 50%, 54.8%, and 57.5%, respectively. In terms of patient health status progress during follow-up, 36.2% and 32.8% improved and maintained progress, respectively. However, in the third and fourth quarters, 79.5%, 77%, and 75.2% of patients had specified treatment during follow-up, knew the symptoms and signs that required a hospital visit, and were given instruction and education during follow-up, respectively.

Figure 1 shows that less than half (42%) of the patients were following up with their health center and one-third (29.75%) were using medical digital technology applications.

Figure 2 shows that 67.2% of the patients had smartphones, 34.8% had medical digital technology applications, and 74.2% had an internet service available all the time. 

Figure 3 shows that 71.75% and 75.5% of the patients used medical digital applications by Sehhaty and Tawakkalna, and 54% and 32.25% used applications by Mawid and Tabeby, respectively.

Table 4 shows that more than half (59.8%) of the patients knew about the health application of the Ministry of Health. Telemedicine consultation was used by 29.8% of the patients and 20% admitted that the use of medical digital technology applications kept them from visiting the hospital. Overall, 80.5% of patients were not provided with all the required medical information, 81.8% had no effective interaction with the medical team about the application, 83.2% had not received any notifications about health instructions through the application, 83.5% did not have medical digital technology applications available at all times and places, and 85.5% could not send medical reports about medical digital technology to the doctor in the hospital.

Figure 4 shows that the majority of the patients did not agree on services provided by medical digital technology applications during the COVID-19 pandemic. Overall, 86.5% of the patients did not agree about medical consultations, 86.5% did not agree about booking an appointment for follow-up in the hospital, 89.5% did not agree about activating the dispensing of prescription drugs, 89.5% did not agree about sending orders for medical analyses, and 89.5% did not agree about sending or exchanging medical reports.

Figure 5 shows that 38.7% of the patients were satisfied by the use of medical digital technology applications for follow-up during the COVID-19 pandemic.

## 4. Discussion 

The COVID-19 pandemic is straining health systems and disrupting the delivery of health-care services, especially for older adults and people with chronic diseases [21]. Digital health is a multidisciplinary field of research that is playing a significant role in boosting traditional health services and developing follow-up for NCDs, especially during COVID-19 [22], via the medical health technologies used for COVID-19 analysis and limiting the current outbreaks of COVID-19 [23].

Findings from this study revealed that less than 10% of the patients in the study were 65 years of age or older. This is similar to Reference [24], in which it is mentioned that less than 10% of patients in the study, on the prevalence and determinants of NCDs in Saudi Arabia, were 65 years of age or more. This highlights the possibility that aging patients are not keen on follow-up due to the aging process.

In the present study, female patients represent less than two-thirds of the sample. This finding does not correspond with Reference [25], a study on NCDs in Saudi Arabia. It was found that the presence of NCDs was higher in males compared to females. This might be because most of the participants who agreed to participate in this study were females. 

In relation to marital status, the results of this study revealed that more than two-thirds of the patients were married. This finding was supported by Reference [24], in which more than two-thirds of the patients were married. This result might be because the age of the patients in this study was 18 years or more.

Regarding patients’ educational attainment, the results of this study revealed that one-quarter of patients were illiterate. This finding was supported by Reference [26], in which one-quarter of Saudi patients with chronic NCDs were illiterate. This finding might be because more than one-third of patients in this study were more than 55 years old, and might not have been concerned with education, or might not have been offered opportunities for further education, in the past.

In the present study, slightly more than 10% of the patients were retired. This finding was supported by Reference [24], in which slightly less than 10% of the patients were retired. This finding might be because less than 10% of the patients in the study were 65 years old or more.

Regarding the area of residence, the findings of this study state that the majority of patients were from the city. This finding does not correspond with Reference [26], in which more than half of patients with NCDs from the south of Saudi Arabia were from rural areas.

Regarding the prevalence of NCDs in Saudi Arabia, more than two-thirds of the patients in this study had diabetes mellitus and more than half had hypertension. This finding does not correspond with Reference [24], in which the prevalence of NCDs in Saudi Arabia was less than one-quarter for hypertension and more than one-tenth for diabetes mellitus. This might be because a large proportion of the patients who participated in this study had diabetes.

The findings of the present study report that two-thirds of patients had a positive family history of NCDs. The same finding was confirmed in another study carried out in Saudi Arabia and given in Reference [27]. The authors of that study confirmed that more than half of the sampled university students in Saudi Arabia had a positive family history of NCDs. This finding might be related to the fact that most NCDs have a hereditary factor.

In this study, the frequency of follow-up was more than one month for more than one-third of the patients. This is compatible with Reference [28], in which more than one-third of the patients with chronic diseases reduced the frequency of hospital visits during the COVID-19 pandemic. This might be related to fear of coronavirus infection.

Regarding progress in health during follow-up, more than 10% of patients in the study had less progress during the COVID-19 pandemic. This finding was consistent with Reference [29], in which less than one-quarter of Saudi patients with chronic disease became worse during the COVID-19 pandemic. This might be because approximately half of the patients in the study did not continue their follow-up during the COVID-19 pandemic.

The current study showed that more than two-thirds of the patients have smart phones, and more than one-third have medical digital technology applications. This result agrees with Reference [30], which reported that the widespread use of mobile technologies can potentially expand the use of telemedicine approaches to facilitate communication between health-care providers. This can increase patients’ access to specialist advice and improve patient health outcomes. Reference [31] reported that the smartphone application tools can provide full consultations when patients are advised not to attend in-person consultations due to the COVID-19 pandemic. 

The current study showed that three-quarters of patients could use an Internet service with constant availability. This result was consistent with Reference [32], which illustrated that digital solutions may have equity implications for at-risk populations with poor Internet access and poor access to digital technology. Telehealth and digital care technologies can benefit society. This study has implications for medical staff, providing information about the potential of digital technologies to provide support during and after the pandemic [21]. This result indicates that individuals depend on the Internet and smartphones for many things, especially since their use at the present time helps protect them against coronavirus infection.

In the present study, more than one-quarter of patients used digital medical applications during the COVID-19 pandemic. This result is in accordance with Reference [33], which reported that more than one-quarter of patients used a telemedicine facility and took telephonic advice from (private) physicians during the COVID-19 pandemic.

Regarding medical digital technology applications used by patients during the COVID-19 pandemic, more than three-quarters of patients in the current study used Tawakkalna, less than three-quarters used Sehhaty, and more than one-half used Mawid. This result agrees with Reference [34], which reported that the Mawid mobile application can be very effective in delivering health-care services in Saudi Arabia during pandemics. This application is one of the health applications provided by the Ministry of Health [35]. The study states that the use of the Tawakkalna application was proven to be successful in fighting the COVID-19 pandemic in the Kingdom of Saudi Arabia.

The researchers’ opinion is that the Ministry of Health provided primary healthcare services during the COVID-19 pandemic for patients. As an example, the infected patients with coronavirus followed up through applications such as Tawakkalna, Sehhaty, Mawid, and Tabaeud. However, other medical digital technology applications are available to monitor NCDs, such as Tabeby online and Seha for medical consultation, but they are less used. This agrees with Reference [36], which concluded that the majority of the patients did not report using any health applications developed by the Saudi Ministry of Health. Reference [37] utilized the newly developed devices designed for home use that facilitated remote monitoring of various physiological parameters relevant to pulmonary disease telemedicine and mobile health technologies. These devices rapidly develop acute care hospital-at-home programs for the treatment of mild-to-moderate cases during the COVID-19 pandemic [38]. The authors of Reference [37] mention that providers working for the adoption of virtual technology for the delivery of medical care may need time to implement changes.

The current study showed that more than half of patients were aware of the health application of the Ministry of Health. This result agrees with Reference [39], which reported that the health-care system is mainly dependent on advanced health technology to cope with the current situation, and it can be a novel tool for improving the health-care system and allowing for better delivery of health-care services during global crises (for example, the COVID-19 pandemic).

The study also showed that less than three-quarters of patients did not use telemedicine consultation, and more than three-quarters did not use medical digital technology applications to prevent hospital visits. This is not in agreement with Reference [39], which reported that technology could be utilized to reduce the burden of both communicable and noncommunicable diseases, as well as to build a patient-centered decision-making health-care system. The majority of patients did not believe that using medical digital technologies provided all of the necessary medical information. There was no effective interaction with the medical team through the application, and no notifications about health instructions through the application were sent. This result disagrees with Reference [40], which reported that eHealth applications may overcome institutional data silos and support holistic and ubiquitous (regional or national) information logistics. Available eHealth indicators mostly describe the usage and acceptance of eHealth in a country. The eHealth indicators focusing on the cross-institutional availability of patient-related information for health-care professionals, patients, and caregivers are rare. This might be because some patients believe that utilization of medical digital technologies is not enough to follow up on their chronic diseases.

This study showed that the majority of the patients reported that medical digital technology applications were not available at all times and places, and that they could not send medical reports through medical digital technologies to the doctor in the hospital. This result contradicts Reference [32], which stated that contact tracing teams reported that digital data entry and management systems were faster to use than paper systems and possibly less prone to data loss [41]; a digital platform that focuses on providing and curating the information used for self-assessment could help physicians make informed changes more accurately for their own clinical practice and decision making; and availability of patient-related data could help physicians make informed changes more accurately for their own clinical practice and decision making [40]. Health-care professionals have full access to patients’ most relevant cross-institutional health record data. This might be because one-quarter of patients in the study were illiterate and approximately another one-quarter could read and write but were not familiar with medical digital technology applications. 

The current study shows that most patients do not agree regarding services provided by medical digital technology applications during the COVID-19 pandemic, including medical consultations, booking an appointment for follow-up in the hospital, activating the dispensing of prescription drugs, sending orders for medical analysis, and sending or exchanging medical reports. These results do not correspond with Reference [42], which reported the positive uptake by both health-care providers and patients after the rapid design and implementation of a digital platform and monitoring services during the COVID-19 pandemic for patients with chronic diseases to support primary health-care services. Reference [43] reported that the Ministry of Health has introduced multiple health applications that provide services such as appointment booking and telemedicine since the beginning of the COVID-19 pandemic. However, we agree with Reference [37] that these applications have been integrated into current clinical workflows and electronic medical records. 

The findings of this study showed that more than one-third of patient satisfaction was related to the medical digital technology applications for follow-up during the COVID-19 pandemic. The results agree with Ref. [30], which reported little evidence of effects on the participants’ health status and well-being, satisfaction, or costs. Reference [42] reported the positive uptake and satisfaction of both health-care providers and patients after the rapid design and implementation of a digital platform.

The researchers’ opinion of the results of the current study is that the services for the Internet, smartphone, and medical digital technology applications presented on the Saudi Ministry of Health website and the mobile application improve health system quality and provide health services for noncommunicable or communicable diseases but are not used effectively. This is due to lack of awareness about these services, with chronic patients being older and having lower levels of education and literacy in relation to technology applications. This agrees with Reference [44], which states that digital health and telemedicine solutions, which exploded during the pandemic, may address many inefficiencies and deficiencies in chronic disease management, such as increasing access to care. However, these solutions are not panaceas and are replete with several limitations, such as low uptake, poor engagement, and low long-term use. 

### Limitations of the Study

This study’s main limitation is as follows: many patients, after being selected by systemic random sampling, refused to participate in the study, so the researcher selected the next patient and excluded them from the sampling. Because clinical outpatient departments only work on certain days of the week, the number of subjects chosen from among outpatients was small. 

## 5. Conclusions

More than one-quarter of the patients in this study used medical digital technology applications. The use of the Tawakkalna and Sehhaty applications was proven to be a successful method in fighting the COVID-19 pandemic in the Kingdom of Saudi Arabia, rather than medical digital technology applications, to follow-up NCDs. Regarding satisfaction, more than one-third of the patients were happy with the services provided by medical digital technology applications for follow-up during the COVID-19 pandemic, so this indicates a need to encourage the use of medical digital technologies. 

### Recommendation 

The study recommends that patients’ awareness of medical digital technology applications and understanding of their utilization and services should be increased through educational programs.

## Figures and Tables

**Figure 1 ijerph-19-12682-f001:**
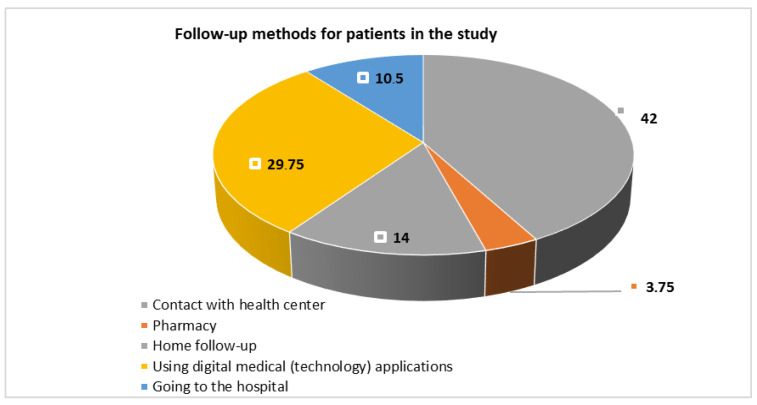
Percentage distribution of the follow-up methods for patients in the study.

**Figure 2 ijerph-19-12682-f002:**
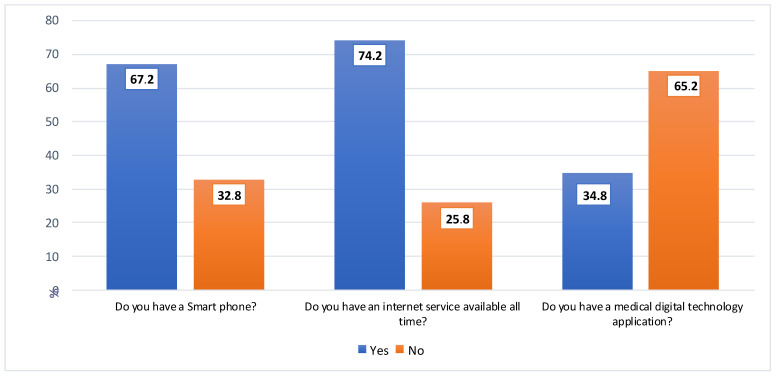
Percentage distribution for the availability of medical digital technologies for patients in the study during the COVID-19 pandemic.

**Figure 3 ijerph-19-12682-f003:**
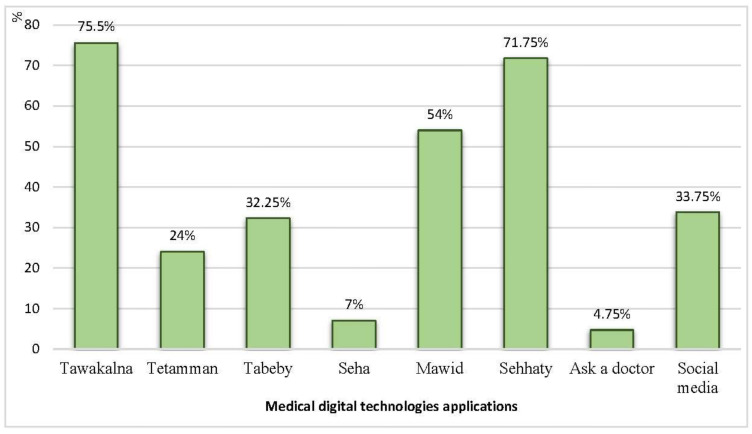
Percentage distribution of medical digital technology applications used by patients in this study during the COVID-19 pandemic.

**Figure 4 ijerph-19-12682-f004:**
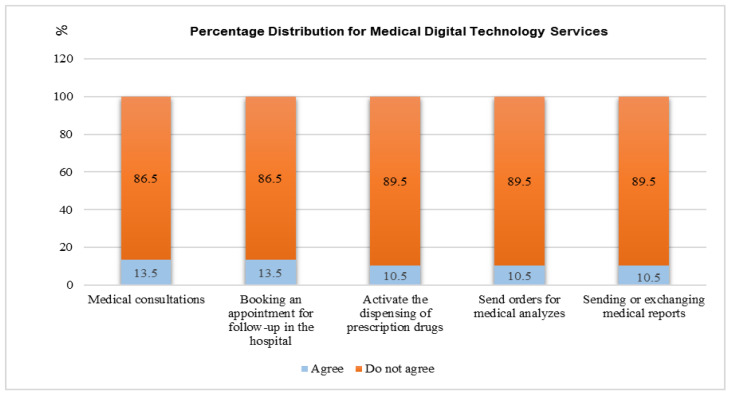
Services of medical digital technologies applications for patients in the study during the COVID-19 pandemic.

**Figure 5 ijerph-19-12682-f005:**
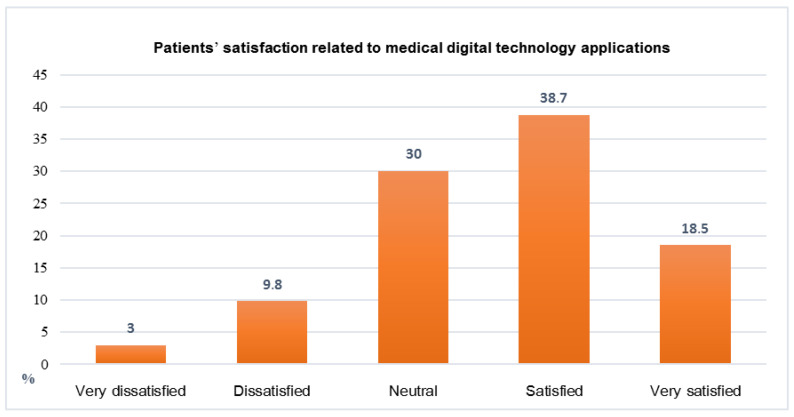
Percentage distribution for patients’ satisfaction related to medical digital technology applications for follow-up during the COVID-19 pandemic.

**Table 1 ijerph-19-12682-t001:** Sociodemographic characteristics for patients in the study.

Items		*N* = 400	%
Age	18 ≥ 25 years	43	10.8
	25 ≥ 35 years	50	12.5
	35 ≥ 45 years	48	12
	45 ≥ 55 years	100	25
	55 ≥ 65 years	136	34
	65 ≥ 75 years	17	4.2
	75 ≥ 85 years	6	1.5
Mean ± SD	47.32 ± 14.362		
Min−max	18–85		
Gender	Male	149	37.2
	Female	251	62.8
Marital status	Single	54	13.5
	Married	283	70.7
	Divorced	19	4.8
	Widow	44	11
Occupation	Worker	9	2.2
	Housewife	126	31.5
	Student	32	8
	Office clerk	110	27.5
	Retired	44	11
	Not working	79	19.8
Educational attainment	Illiterate	100	25
	Literate	94	23.6
	Secondary level	85	21.2
	Highly qualified	121	30.2
Financial status	Not enough to meet basic and medical needs	115	28.8
	Enough to meet basic and medical needs	285	71.2
Residence (type of town)	Village	64	16
	City	336	84

**Table 2 ijerph-19-12682-t002:** Health history of patients in the study.

Items.		*N* = 400	%
* Patient diagnosed with:			
▪ Hypertension		230	57.5
▪ Diabetes mellitus		277	69.2
▪ Chronic respiratory diseases		37	9.25
▪ Heart disease		35	8.75
▪ Hypotension		2	0.5
▪ Cancer		1	0.25
Duration of disease	▪ <1 year	32	8
	▪ 1–5 years	100	25
	▪ 5–10 years	138	34.5
	▪ >10 years	130	32.5
Regular medication	▪ Yes	326	81.5
	▪ No	74	18.5
Family history for noncommunicable disease	▪ Yes	264	66
	▪ No	136	34
Smoking	▪ Yes	97	24.2
	▪ No	303	75.8
Previous accidents	▪ Yes	74	18.5
	▪ No	326	81.5
Allergic conditions	▪ Yes	127	31.8
	▪ No	273	68.2
Sufficient time to sleep	▪ Yes	254	63.5
	▪ No	146	36.5
Passive smoking	▪ Exposed	182	45.5
	▪ Not exposed	218	54.5

* Answers are not mutually exclusive.

**Table 3 ijerph-19-12682-t003:** Follow-up characteristics for patients in the study.

Items	*N* = 400	%
Frequency of follow-up:		
▪ First time	31	7.8
▪ Weekly	20	5
▪ Monthly	210	52.4
▪ More than a month	139	34.8
Pattern of follow-up:		
▪ Emergency cases	96	24
▪ Take monthly medication	200	50
▪ Routine system for checkup	104	26
Progress of health status during follow-up:		
▪ Improvement	145	36.2
▪ Maintained progress	131	32.8
▪ Less progress	57	14.2
▪ General checkup/maintenance	67	16.8
* Desire to visit member of the health team during follow-up:		
▪ Specialist doctor	368	92
▪ Specialist nurse	102	25.5
▪ Lab	29	7.2
▪ Pharmacy	93	23.2
Going to follow-up with a relative:		
▪ Yes	181	45.2
▪ No	219	54.8
Specified treatment during follow-up:		
▪ Yes	318	79.5
▪ No	82	20.5
Symptoms and signs known to the patient requiring a visit to the hospital:		
▪ Yes	308	77
▪ No	92	23
Instructions/education given during follow-up:		
▪ Yes	301	75.2
▪ No	99	24.8
Any change in medication during follow-up:		
▪ Yes	230	57.5
▪ No	170	42.5

* Answers are not mutually exclusive.

**Table 4 ijerph-19-12682-t004:** Data about applications of medical digital technologies for patients in the study during the COVID-19 pandemic.

Items		*N* = 400	%
Do you know the health applications of the Ministry of Health?	Yes	239	59.8
	No	161	40.2
Do these applications use social distancing during the COVID-19 pandemic?	Yes	172	43
	No	228	57
Do you use telemedicine consultations?	Yes	119	29.8
	No	281	70.2
Does the use of medical digital technology applications prevent you from visiting the hospital?	Yes	80	20
	No	320	80
Do digital medical technology applications provide you with all the medical information you need?	Yes	78	19.5
	No	322	80.5
Is there an effective interaction with the medical team through the application?	Yes	73	18.2
	No	327	81.8
Does it send you notifications of health instructions through the application used?	Yes	67	16.8
	No	333	83.2
Are medical digital technology applications available at any time and place?	Yes	66	16.5
	No	334	83.5
Can you send medical reports through medical digital technology to the doctor in the hospital?	Yes	58	14.5
	No	342	85.5

## Data Availability

The datasets used and analyzed during the current study are available from the corresponding author on reasonable request.

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
