# Peer review of "Applications of Medical Digital Technologies for Noncommunicable Diseases for Follow-Up during the COVID-19 Pandemic"

_ijerph, 2022, doi:10.3390/ijerph191912682_

Round 1
Reviewer 1 Report
MAIN POINTS:
- Article needs to be edited thoroughly for language, grammar, and formatting.
- Past tense language needs to be implemented throughout.
- Patient-centric language needs to be checked throughout.
- Period of "during COVID-19 pandemic" needs to be defined. WE are technically not out just yet.
Abstract:
- Line 3: ... NCD's among who exactly?
- Line 5: This study aimed to explore the application of medical technology for patients with NCD's during... hospitalization? brief visitation? COVID diagnosis? and following release from hospital? This time need to be clearly defined.
- Line 6: Methods sections should always start with participants.
- What does systematic random sampling mean exactly?
- Line 11: "the period of COVID-19" needs to be deciphered further. The pandemic has not been declared over just yet. Use concrete dates.
- For the abstract, you could generalize the technological platforms as just that. It can be explained further in the body of the text.
INTRODUCTION:
- "The prevalence and impact of noncommunicable..." Impact of what exactly?
- Line 6: risk factors ... for disease? What exactly are you referring to here?
- Paragraph (lines 25-27) is disjointed. Merge it with others.
- Line 30: lockdowns need to be referred to as stay at home orders. This term does not look the same globally. Social distancing should be referred to as physical distancing.
Line 31: ... countries may specifically affect...
Line 37: ... an essential component of and an enabler...
Line 39: What are the broader health needs you are referring to?
MAGNITUDE OF THE PROBLEM
- Line 1: What are COVID-19 response actions? Are you referring to public health responses to COVID-19? ...on NCD's or patients with NCD's?
- Line 3: What exactly do you mean by unhealthy diet? What does this look like?
- Line 7: Social distancing should be reported as physical distancing.
- Line 8: What exactly is the rationale for this study? How does this tie in to the impacts of NCD's on patients during COVID? This connections needs to be clarified further.
AIM OF THE STUDY
- This study was aimed ... exploration of (usage?) of medical digital technologies among (patients?) with NCD's following (release from the hospital?) during the dates of ( date to date).
RESEARCH QUESTION
- The question doesn't make sense. This needs to be rewritten to more clearly define what exactly you have aimed to investigate.
SUBJECTS AND METHODS
- Use past tense language.
- You need to be more descriptive. For example, The participants of this study were 400 adults. Men (n = # age; +/- SD) and women (n = # age; +/- SD) enrolled in this study all were diagnosed with an NCD and were (in patient? or outpatient?). What does "critical patient" mean? Current or past alcohol consumption? History of drug abuse? These need to be clarified in greater detail.
SAMPLING DESIGN
- What is systematically randomly selected?
- What does the n number of 378 mean?
STRUCTURED INTERVIEW
- What does location refer to exactly?
- Educational level should be educational attainment
NCD QUESTIONNAIRE
- What does simple Arabic language mean exactly?
- What are the differences between the first questionnaire and the structured questionnaire? There should be some explanation as to why there are two different instruments used.
FIELDWORK
- Ten to fifteen should be ten-to-fifteen
- Excel sheet should be excel spreadsheet
VALIDITY AND RELIABILITY
- Who are the 5 experts? Nurses? How are nurses trained or capable of performing an expert validation of this?
ETHICAL APPROVAL
- ... remove 'unconditional.' You received approval, that is all that is necessary.
- Check spelling in all tables. Each of the categories should have a note to explain what the categories mean. For example, middle certification, sufficient for basic and medical needs, etc... "Divorce" is divorced.
-Check all table notes for spelling and grammar. There are many instances of errors.
- Figures need error bars, standard deviations, and significance. Notes need to explain these numbers are well. All axes need to be labeled.
- What does "under the study mean?"
- Figure title need to be checked for punctuation
DISCUSSION
- How is COVID-19 straining health systems exactly? This needs explanation. Is this global? Localized? The discussion is very lengthy and could use some referring to table or figure information. Data needs to be included in your discussion, but much of what is written should be in the results section. It is far too drawn out in length. Reduce and be more concise.
CONCLUSION
- This reads more like what the discussion should be. Your conclusion should be the boiled down and reduced (1-2 sentence) statement to reflect your findings.
AUTHORS CONTRIBUTIONS
- Check for spelling, grammar, and punctuation.
Author Response
Dear Editor,
Thank you for your interest and very great reviewers. I am deeply grateful to them. I am indebted to their constructive criticism, expertise, and unlimited help. I appreciate their active participation in providing me with a lot of knowledge. I did the schedule to clarify the comments on how it was done according to reviewers 1 and 2.
Response to Reviewer 1 Comments
|

Reviewer 2 Report
The proposed paper ‘Applications of Medical Digital Technologies for Noncommunicable Diseases for Follow-up During COVID-19 Pandemic’ analyses the results of self-compiled questionnaires submitted to 400 patients in the territory of Saudi Arabia. The study reports the description of the subject population, and their level of awareness, use and satisfaction about m-Health applications for clinical follow-up, with a specific focus on the COVID-19 pandemic scenario. While the method is very traditional and doesn’t present any innovation, the specific context of the study makes the results interesting and valuable. Some issues should in my opinion be addressed before publication.
MAIN ISSUES:
1.1) The language of the manuscript requires extensive review, especially in section 4 (discussion), where the reading and understanding of the text is often hard due to the questionable grammar.
1.2) In section 4 (discussion), a paragraph for limits and future developments should be added, addressing the main limitations to the validity of the study (mainly the low number of subjects involved) and possible ways to tackle them.
MINOR ISSUES:
Section 1
2.1.1) ‘The safest mode of follow-up for the delivery of medical services is virtual or telecare.’ What do authors mean by ‘safe’? It is a statement that requires a reference. If it is intended as safe in a strictly ‘covid-wise’ perspective, it must be specified.
2.1.2) ‘the importance of linking medical services these strategic priorities has been demonstrated’ Unclear sentence, should be rephrased.
2.1.3) Research question:
‘What about medical digital technologies applications for patients with NCDs follow-up during COVID-19 pandemic?’ In my opinion the question is posed in too generic terms (‘what about’), I suggest to either elaborate it, and specify a more precise focus, or to remove this paragraph.
Section 2
2.2.1) Research design: I think that, for clarity sake, it should be slightly elaborated, explaining that it is a statistical study based on self-compiled questionnaires.
2.2.2) The graphical use of titles, subtitles and list points (e.g. Tools of Data Collection) is a little confusing, I suggest to rework it.
2.2.3) I could not find the timing of the research reported in the text. When were the questionnaires submitted to the subjects? How long did the process take?
Section 3
2.3.1) Grey and green colors in figure 1 are very similar and quite hard to distinguish in the legend, I suggest to revise it.
2.3.2) Figure 3 and 4: specify that the Y axis is reporting a percentage value for a relative frequency.
2.3.3) I don’t understand the meaning of what is reported in figure 4. What do authors mean by ‘agreement’ from the patients? Does it mean that the patients can or cannot find this functionality in the used application, that they do or do not want to use it, that they do or do not want them to be included, that they are satisfied or not, or something else?
Section 4
2.4.1) ‘This might be because a large proportion of the patients who participated in this study had diabetes.’ This appears to be tautological and therefore not interesting. I suggest defining this result as simply due to randomness in the selection process and to the limited sample size.
2.4.2) ‘The use of the Tawakkalna application was proven to be a successful method in fighting COVID-19 pandemic in the Kingdom of Saudi Arabia.’ A reference is necessary.
2.4.3) ‘The researchers’ opinion that the primary health-care services during COVID-19 pandemic for patients with chronic diseases, which refer to follow up the coronavirus pandemic and consequence the infected patient through the digital application such as Tawakkalna, Sehhaty, Mawid and Tabaeud relate to Ministry of Health, while medical digital technology application to follow-up the NCDs as Tabeby online, Tabeby for medical consultation online, and others services less used.’ I was not able to understand this paragraph, please rephrase it. This is one example of what was addressed at point 1.1.
Author Response
Dear Editor,
Thank you for your interest and very great reviewers. I am deeply grateful to them. I am indebted to their constructive criticism, expertise, and unlimited help. I appreciate their active participation in providing me with a lot of knowledge. I did the schedule to clarify the comments on how it was done according to reviewers 2.
Response to Reviewer 2 Comments
|